# Unraveling the Role of Ras Homolog Enriched in Brain (Rheb1 and Rheb2): Bridging Neuronal Dynamics and Cancer Pathogenesis through Mechanistic Target of Rapamycin Signaling

**DOI:** 10.3390/ijms25031489

**Published:** 2024-01-25

**Authors:** Mostafizur Rahman, Tuan Minh Nguyen, Gi Jeong Lee, Boram Kim, Mi Kyung Park, Chang Hoon Lee

**Affiliations:** 1College of Pharmacy, Dongguk University, Seoul 04620, Republic of Korea; mostafizur@dgu.ac.kr (M.R.); zxcvb2736@naver.com (G.J.L.);; 2Department of BioHealthcare, Hwasung Medi-Science University, Hwaseong-si 18274, Republic of Korea

**Keywords:** Rheb1, Rheb2, cancer, mTOR, hallmarks, neurons, Rheb1 inhibitors

## Abstract

Ras homolog enriched in brain (Rheb1 and Rheb2), small GTPases, play a crucial role in regulating neuronal activity and have gained attention for their implications in cancer development, particularly in breast cancer. This study delves into the intricate connection between the multifaceted functions of Rheb1 in neurons and cancer, with a specific focus on the mTOR pathway. It aims to elucidate Rheb1’s involvement in pivotal cellular processes such as proliferation, apoptosis resistance, migration, invasion, metastasis, and inflammatory responses while acknowledging that Rheb2 has not been extensively studied. Despite the recognized associations, a comprehensive understanding of the intricate interplay between Rheb1 and Rheb2 and their roles in both nerve and cancer remains elusive. This review consolidates current knowledge regarding the impact of Rheb1 on cancer hallmarks and explores the potential of Rheb1 as a therapeutic target in cancer treatment. It emphasizes the necessity for a deeper comprehension of the molecular mechanisms underlying Rheb1-mediated oncogenic processes, underscoring the existing gaps in our understanding. Additionally, the review highlights the exploration of Rheb1 inhibitors as a promising avenue for cancer therapy. By shedding light on the complicated roles between Rheb1/Rheb2 and cancer, this study provides valuable insights to the scientific community. These insights are instrumental in guiding the identification of novel targets and advancing the development of effective therapeutic strategies for treating cancer.

## 1. Introduction

Ras homolog abundant in brain (Rheb1) and Rheb2 belong to the Ras superfamily of small GTPases, found from yeast to mammals but absent in prokaryotes [1,2,3,4]. Both Rheb1 and Rheb2 proteins function as monomeric switches, pivotal in various biological processes. The activity of Rheb1 holds critical importance in both health and disease contexts [5,6,7]. TSC/Rheb1/mTOR (mammalian target of rapamycin) signaling governs protein synthesis and growth, responding to factors like food, energy, and growth circumstances. Rheb1 plays a central role within this signaling pathway [8,9,10,11,12]. Particularly in tuberous sclerosis complex (TSC), a hereditary condition linked to seizures, mental retardation, and the development of benign tumors (hematomas) in diverse body regions such as the brain, kidney, lung, and skin, Rheb1’s significance is noteworthy [13,14]. Rheb1’s implications stretch across various disorders, including cancer, diabetes, aging, neurodegeneration, epilepsy, and autism, all involving excessive mTORC1 signaling [15].

In this review, we will summarize the normal functions of Rheb1 and Rheb2, their roles in cancer development, potential pathways for Rheb1 or Rheb2-mediated carcinogenesis, and explore possible treatment options. We anticipate that this review will provide a robust foundation for new research into Rheb1 and Rheb2 as potential targets for cancer therapy. If information on Rheb2 is unavailable, we will focus solely on describing Rheb1, given its extensively reported role, acknowledging the limited studies on Rheb2.

## 2. Biology of Rheb1 and Rheb2

In 1994, Yamagata et al. discovered the *Rheb1* gene [3]. Rheb1 proteins, with a molecular weight of approximately 21 kDa, exist as monomeric proteins found in organisms ranging from yeast to humans. Notably, plants do not possess Rheb1 [5,6,7]. Initially identified as a gene swiftly triggered in the brain by convulsions and receptor-dependent synaptic activity, Rheb1 is ubiquitously expressed [3,16]. Homologs of the *Rheb1* gene are present in various non-mammalian and mammalian species, including mice, rats, and humans [2,3,17]. While lower eukaryotes like yeast or Drosophila have a single gene, humans harbor two distinct genes: *hRheb1* and *hRheb2*, also known as *RhebL1*. The human *Rheb1* gene resides on chromosome 7 (7q36), while its paralogous counterpart sits on chromosome 12 (12q13.12), encoding Rheb2 [1]. The genomic structure of human Rheb1 comprises seven exons, whereas human Rheb2 consists of eight exons.

Rheb1 exhibits ubiquitous expression, with pronounced levels in skeletal and cardiac muscles, while Rheb2 expression is more confined, showing higher levels in the brain [18]. Additionally, Rheb1 and Rheb2 are localized at endomembranes [19]. Studies in MDCK, HeLa, astrocytoma, and HEK293 cells revealed that Rheb1 and Rheb2 induce substantial cytoplasmic vacuoles, characterized as late endocytic components (resembling late endosomes and lysosomes) [18]. This vacuole formation necessitates the GTP form of Rheb but does not rely on the activation of the downstream mTOR kinase. This suggests that Rheb regulates the endocytic trafficking pathway independently of the previously identified mTOR pathway [18]. In COS-7 cells, both Rheb1 and Rheb2 localize to ER and Golgi but not to endosomes, lysosomes, or mitochondria [20]. Rheb2 also activates mTORC1 in the in vitro system [21]. Moreover, *hRheb2* and *hRheb1* can compensate for the role of *S. pombe Rheb* [22], indicating that Rheb2 might play a significant biological role similar to Rheb1.

### 2.1. Structure of Rheb1 and Rheb2

In humans, Rheb1 is composed of 184 amino acids, while Rheb2 consists of 183 amino acids, sharing a 51% identity [23]. Both proteins exhibit a GTPase domain at the N-terminus and a highly variable region at the C-terminus, concluding with a CAAX motif. Crystal structures of Rheb1 bound to GTP (guanosine triphosphate), GppNHp (guanosine-5′-(βγ-imino) triphosphate), and GDP (guanosine diphosphate) have been determined, highlighting their structural differences [24] (Figure 1).

Rheb1’s structure displays notable similarities to other small GTPases, showing a closer structural relationship to Ras and Rap compared to Rab5A and RhoA [24]. These proteins bind guanine nucleotides, transitioning between GTP- and GDP-bound states. GTPase activator proteins (GAPs) [25,26] maintain these proteins in an inactive GDP-bound state, while guanine nucleotide exchange factors (GEFs) [27] keep them active in a GTP-bound state [28,29].

During the GDP/GTP cycle, Rheb1’s switch I region undergoes conformational changes, while the switch II region, unlike other GTPases, maintains a relatively stable structure. Rheb1’s switch II region assumes a unique conformation distinct from the long-helical shape observed in other Ras family GTPases. Additionally, due to Rheb1’s distinctive structure, Gln64 (equivalent to Gln61 in Ras) is embedded in a hydrophobic core, hindering its interaction with GTP or the catalytic active site. Another differentiation from Ras is the shielding of the phosphate moiety of GTP by a conserved tyrosine residue, Tyr35, in Rheb1 (comparable to Tyr32 in Ras).

These distinct structural traits in Rheb1 indicate a different GTPase mechanism from that of Ras. Studies by Mazhab-Jafari et al. [30] revealed that Tyr35 of Rheb1 inhibits its intrinsic GTPase activity, maintaining it in a highly activated state. Conversion of this residue to alanine (Y35A) results in a tenfold increase in intrinsic GTPase activity. Further analysis of the mutant suggests that GTP hydrolysis is facilitated by Asp65 in switch II and Thr38 in switch I. Tyr35 seems to restrict the conformation of the active site, preventing Asp65 from accessing the nucleotide-binding pocket. Notably, the TSC2 GAP activity does not affect the Y35A mutant.

Analysis of residues 1–169 of Rheb1 expressed in *E. coli* yielded NMR spectra of Rheb1 [31]. The HSQC spectra of GDP- and GppNHp-bound Rheb1 were examined using 1H-15N heteronuclear single-quantum coherence (HSQC) spectroscopy, revealing distinct chemical shifts in residues around the P-loop (residues 10–20) and the switch II region (residues 60–65). A separate investigation determined the structure of Rheb1-GDP using NMR [32].

The CAAX motif (where A is an aliphatic amino acid and X is a C-terminal amino acid) is situated in the carboxyterminal hypervariable region (HVR) of Ras family GTPases. Both Rheb1 and Rheb2 possess this CAAX motif, but their HVR lacks palmitoylated cysteines and a polybasic domain. Farnesylation of Rheb1 and Rheb2 occurs at the cysteine, followed by Rce1 cleavage of the AAX motif and subsequent carboxyl methylation by Icmt1 [20,33]. These post-translational modifications are necessary for Rheb1’s attachment to endomembranes [33]. The absence of palmitoylated cysteines or a polybasic domain might explain why Rheb1 is not associated with the plasma membrane. Interestingly, when Rheb1’s HVR was replaced with the H-Ras HVR, Rheb1 was observed to localize to the plasma membrane [33].

### 2.2. Rheb1 Expression

Initially, Rheb1 was identified as a gene that is highly expressed in rat and adult human tissues, with the highest levels found in skeletal and cardiac muscle [1]. In human carcinomas, Rheb1 is typically overexpressed [34]. Meta-analysis of cancer cytogenetic and transcriptional datasets unveiled frequent amplification of the Rheb1 gene on chromosomes 7q36.1–q36.3 across various human cancer histologies. Notably, higher Rheb1 expression has been observed in liver, lung, and bladder malignancies. Analysis of the Microarray Database revealed a significant correlation between elevated Rheb1 mRNA expression and breast cancer progression. Further evidence for Rheb1’s role in neoplastic transformation comes from experiments on chicken embryonic fibroblasts expressing Rheb1 mutants [35]. The expression of Rheb1 Q64L and N153T induced morphological changes in cells, increasing their size, inducing vacuolization, and enabling anchorage-independent growth.

Under different conditions, Rheb1 expression is upregulated. Treatment with TGFβ1 significantly induced mRNA and protein expression of Rheb1 in NRK-49F cells [36]. Studies exploring the host inflammatory response via miRNA155 (miP155) revealed its regulatory role in Rheb1 expression [37]. miP155 binds to the 3′ untranslated region of Rheb1, linking it to mycobacterial clearance during *Mycobacterium tuberculosis* infection and bacterial clearance in Pseudomonas aeruginosa keratitis during corneal infection [38]. In immortalized human hepatocytes, Rheb1 expression increases upon HCV infection [39]. Moreover, hydrogen peroxide induces ubiquitination and degradation of Rheb1 in GSH-depleted RAW 264.7 cells, leading to Beclin1-independent autophagic cell death [40].

### 2.3. Mutation of Rheb1

Rheb1 mutations have been categorized into several forms, primarily divided into activating mutants, loss-of-function mutants, and membrane association mutants. The identification of Rheb1 mutations in the human cancer genome database led to the discovery of certain activating mutants [41]. Table 1 enumerates confirmed Rheb1 mutations known to impact its function.

A structure-based investigation highlighted Q64L as an extensively used activating mutation with enhanced GTP loading and partial resistance to TSC-GAP [45]. Recent discoveries have identified a mutant displaying even more pronounced activation [44]. Structural analyses of Rheb1 emphasized the significance of the G3-box DxxG motif in coordinating a Mg^2+^ ion, crucial for aligning the catalytic H_2_O near the glycine-C-phosphate. This understanding led to substitutions impacting intrinsic GTP hydrolysis [44]. The resulting G63A mutant showcased reduced intrinsic and TSC2-GAP-mediated GTPase activity, boosting mTORC1 signaling activation relative to wild-type Rheb1, as demonstrated by S6K phosphorylation.

Some activating mutations surfaced through mutagenesis experiments. Urano et al. screened a mutant Rheb1 library in *S. pombe* to uncover hyperactivating mutants of fission yeast Rheb1 [46]. These mutations, when introduced into human Rheb1, unveiled heightened mTORC1 signaling, notably in the N153S mutant and, to a lesser extent, the S89D mutant [42]. Yan et al. [42] also identified two significantly activating mutations, S16N and S16H. The discovery of activating Rheb1 mutants stemmed from investigations into human cancer genome databases. A comprehensive genomic study of 4742 human cancer samples revealed 33 genes previously unidentified as altered in cancer, including genes linked to cell growth [49]. Rheb1 emerged as one of these genes, with the Y35N mutation detected in five tumors (two endometrial and three renal clear cell cancers). The expression of Rheb1 Y35N/C/H mutants heightened S6K phosphorylation [43]. Furthermore, expression of the E139K mutant Rheb1 resulted in a minor increase in phospho-S6K.

Loss-of-function mutants were discovered by altering residues in the effector domain (switch I). Sato et al. [21] performed an in vitro mTORC1 activation assay to assess the impact of altering effector domain residues. The mutations D36A, P37A, T38A, and N51A were found to hinder Rheb1’s ability to activate mTORC1. The objective was to identify crucial areas for mTORC1 signaling by mutating 65 residues across Rheb1’s entire surface to alanine [48]. These studies indicated that mutations Y67A/I69A and I76A/D77A result in loss of function, intriguingly located in the switch II segment. Surprisingly, extensive alterations of residues in the switch I region had minimal effects on Rheb1 function. This suggests that specific residues in switch II play a role in downstream effector interaction. Mutagenesis investigations also revealed dominant negative Rheb1 mutants (S20N and D60I/K/V) that diminish mTOR signaling [12,47]. However, the low expression of these dominant negative mutants limits their application in signaling research.

### 2.4. Rheb1 and Rheb2 Signaling Pathways

Growth hormones, such as insulin-like growth factor 1 (IGF-1) and insulin, stimulate the lipid kinase phosphatidylinositol-3 kinase (PI3K) through receptor tyrosine kinases or G-protein-coupled receptors [50,51]. Akt, a serine/threonine kinase, stands as one of the key downstream mediators of PI3K signaling [47,50,51,52,53]. Akt-mTOR signaling involves the TSC complex protein and Rheb1 GTPase. The TSC complex, comprising TSC1 (hamartin), TSC2 (tuberin), and TBC1D7, acts as a GAP (GTPase-activating protein) towards Rheb1, suppressing its activity [1,45,52,54,55,56,57,58,59,60,61,62,63]. Akt inhibits TSC2 GAP activity by phosphorylating it at conserved consensus phosphorylation sites in vitro [59,60]. Reduced GAP activity of the TSC complex leads to the accumulation of GTP-bound Rheb1 over GDP-bound Rheb1 [45,58,63,64]. Compared to other Ras-related small G proteins, Rheb1 exhibits low intrinsic GTPase activity [60,65]. Consequently, Rheb1’s GTP/GDP loading state is tightly regulated by its GAP, influenced by the presence of growth factors. Rheb1 binds to its effector, mTOR, where GTP-loaded Rheb1 is essential for mTOR activation [47,51,59,66] (Figure 2). The biochemical and physiological significance of a GEF (guanine nucleotide exchange factor) for Rheb1 remains to be determined [6,65]. Elevated expression of the Rheb2 transgene results in heightened mTORC1 activity in HepG2 cells [67].

mTOR, an evolutionarily conserved checkpoint protein kinase, has emerged as a pivotal regulator of cell growth and proliferation. Triggered by secreted growth factors and influenced by cellular levels of amino acids, glucose, energy, and oxygen, the mTOR pathway governs protein synthesis, autophagy, and metabolism [52,60,68,69,70]. The mTOR complexes, mTORC1 and mTORC2, function independently [71]. mTORC1 consists of mTOR, Raptor, mammalian lethal with SEC13 protein 8 (mLST8), along with two endogenous inhibitors—PRAS40 and DEP domain-containing mTOR-interacting protein (DEPTOR) [47,52,68,71,72]. Activated mTORC1 aids protein translation through phosphorylation of 70 kDa S6K and eukaryotic translation initiation factor 4E-binding protein 1 (4E-BP1) while reducing autophagy through Ulk1/Atg13 [52,73,74,75,76,77]. Rapamycin, an immunosuppressant and anticancer drug, binds strongly to the 12-kDa FK506-binding protein (FKBP12), inhibiting mTORC1 function and thus cell proliferation [52,68,71,78,79]. Apart from mTOR and mLST8, mTORC2 includes crucial polypeptides—Rictor (mTOR’s rapamycin-insensitive sidekick) and mammalian stress-activated map kinase-interacting protein 1 (mSin1) [68,71,80]. mTORC2 potentially interacts with the actin cytoskeleton via Rho and Rac [68,71,80,81,82].

### 2.5. Binding Partners of Rheb1

Recently, various TSC complex and Rheb1 signaling pathways independent of mTORC1 have been uncovered [81,83,84,85] (Table 2). Activation of these non-canonical pathways contributes to certain clinical symptoms of TSC that resist rapamycin-induced suppression of mTORC1 [86]. Investigations into small compounds targeting Rheb1 have aimed to broaden therapeutic possibilities. Recent research has identified 4,4′-biphenol as a compound binding to Rheb1 and impacting cell viability [65,87].

Beyond mTORC1, Rheb1 exhibits several non-canonical functions, including reducing aggresome formation [84], modulating endocytic trafficking pathways [18], and influencing cellular fate through notch signaling [88]. Mediators of Rheb1’s non-canonical signaling include B-Raf kinase [89,90]; FKBP38 [91]; NIX, LC3, [92] and RASSF1A [93]; carbamoyl-phosphate synthetase 2, aspartate transcarbamoylase, and dihydroorotase (CAD) [94,95]; β-site amyloid precursor protein (APP)-cleaving enzyme 1 (BACE1) [96]; protein kinase-like endoplasmic reticulum kinase (PERK) [97]; and syntenin [95].

**Table 2 ijms-25-01489-t002:** The list of proteins that have been reported to interact with Rheb1.

Proteins Involved in Canonical Signaling Pathway	
TSC complex	In the exclusion of growth hormones or insulin, the TSC complex increases Rheb1’s intrinsic GTPase activity on the lysosomal surface and forms a complex with Rheb1 at the lysosomal membranes by converting Rheb1-GDP.
Bnip3	Bnip3 interacts with Rheb1 in hypoxia and suppresses its function by preventing Rheb1 contact with downstream targets or interfering with Rheb1 GTP loading [98].
mTOR	Rheb1 binding to mTOR is independent of its GTP-bound state, yet a GTP-bound state is required for mTORC1 activation, boosting growth, cell cycle advancement, and autophagy suppression.
GAPDH	GAPDH interacts with Rheb1 in the absence of glucose, regardless of Rheb1’s guanyl nucleotide-loaded condition.
PLD1	In a GTP-dependent way, Rheb1 binds to and activates phospholipase D1 (PLD1), and Rheb1 stimulates PLD1 to create phosphatidic acid, which activates mTORC1 indirectly [99].
**Proteins Involved in Non-Canonical Signaling Pathway**	
RASSF1	By limiting mTORC1 signaling, RASSF1 interacts with Rheb1 to promote autophagy.
LC3	Rheb1 physically interacts with the mitochondrial autophagic receptor Nix and the autophagosomal protein LC3-II. The recruitment of Rheb1 to mitochondria leads to the activation of mitophagy.
NIX	Rheb1 is recruited to the mitochondrial outer membrane in response to strong oxidative phosphorylation activity. Rheb1 stimulates mitophagy by interacting physically with Nix, the mitochondrial autophagic receptor, and LC3-II, the autophagosomal protein.
Syntenin	Rheb1-GDP regulates spine development by binding syntenin.
FKBP38	FKBP38 belongs to the FK506-binding protein family and functions as a mTOR antagonist [100]. Rheb1, which interacts with FKBP38 in a GTP-dependent way and hinders it from interacting with mTOR, counteracts the inhibitory effect of FKBP38. Rheb1 and the anti-apoptotic protein Bcl-2 attach to the same area on FKBP38; therefore, Rheb1 displaces Bcl-2, allowing Bcl-2 to interact with pro-apoptotic proteins [91].
PERK	Rheb1 forms a GTP-dependent association with PERK to limit protein synthesis via eIF2 signaling during ER stress.
CAD	Rheb1 regulates pyrimidine synthesis via binding to CAD at its C-terminal carbamoyl phosphate synthetase domain in a GTP-dependent manner. This connection is dependent on Rheb1’s effector domain.

## 3. Role of Rheb1 and Rheb2 in Neurons

Extensive studies on Rheb1 have focused on its role in neurons. Initially, Rheb1 was identified as an immediate early response protein whose expression surged due to NMDA (N-methyl-D-aspartate)-dependent synaptic activity in the brain [101] (Figure 3). While initially studied predominantly in neurological disorders, its significance in cancer has recently gained recognition.

### 3.1. Rheb1 in Neural Stem Cell Differentiation

In many mammalian tissues, stem cells persist throughout life, facilitating the replacement of lost cells in disease and maintaining tissue health during regular turnover. The activation of mTORC1 by Rheb1 is essential for the development and differentiation of neural stem cells (NSCs) or neural progenitor cells (NPCs) during embryonic development [101]. Knocking off Rheb1 in NSCs in mice leads to postnatal growth failure, weight loss, and eventually death [102]. NPCs, crucial for generating neurons and glia during development, rely on mTORC1’s ability to phosphorylate the transcription factor Stat3, which is pivotal for glial differentiation [101]. NSC self-renewal and differentiation are vital processes at various stages of brain development. Overactive mTORC1 encourages NSC differentiation at the expense of self-renewal in neonatal NSCs, while reduced mTORC1 activity prevents differentiation, resulting in fewer neurons [103]. Additionally, due to its involvement in Reelin-Dab signaling, constant activation of Rheb1 in NPCs influences neuroblast migration [104]. While Rheb1 is necessary for NPC differentiation, its inappropriate activation can lead to improper differentiation.

### 3.2. Rheb1 and Rheb2 in Neuronal Growth and Energy

Silencing Rheb1 in embryonic mice proves lethal, while in adult mice, it leads to a shortened lifespan [105]. These findings collectively highlight Rheb1’s essential role in the survival of both embryonic and adult animals [105]. In Drosophila’s visual system, Rheb1 governs differentiation independently of cell cycle and cell size regulation [106]. These discoveries indicate that the activation of the insulin receptor-mTOR pathway significantly regulates the timing of neuronal differentiation [106]. Notch signaling is crucial in maintaining stem/progenitor cells in an undifferentiated state and plays a pivotal role in regulating NPC development. Rheb1, operating independently of mTORC1, influences cell fate in the Drosophila external sensory organ (ESO) through the notch pathway [88]. Activation of Rheb1 induces defects in asymmetric cell division in Drosophila sensory organ precursor cells, an intriguing observation [88]. In the mammalian system, insulin signaling is also vital for age-dependent neuronal development and differentiation [107,108]. Constitutive activation of Rheb1 in mouse NPCs and neuroblasts in the subventricular zone may lead to premature neuronal differentiation and disruption of cell migration pathways, hindering migration or releasing abnormal signals [108]. Elevated Cul5 expression caused by hyperactivated mTORC1 disrupted Reelin-Dab1 signaling, resulting in abnormal neuronal migration in TSC neurons [104]. Both hyperactivation and inhibition of the Rheb1-mTORC1 pathway impair differentiation and can lead to premature death, as observed in studies using TSC and Raptor knockout mice [104,109]. A reduction in mTORC1 activity in specific areas of the adult brain affects learning, memory, and social behavior [105,109]. Notably, in the mouse brain, deletion of Rheb2 did not affect the levels of pS6, pAKT S473, or any other components in the mTOR pathway, indicating that while Rheb1 is vital for mTORC1 signaling and embryonic survival, Rheb2 is not essential for either embryonic survival or mTORC1 signaling [110].

In mouse brain and liver tissue, Rheb1 controls the flow of acetyl-CoA in the mitochondrial tricarboxylic acid cycle by stimulating pyruvate dehydrogenase (PDH), leading to an augmentation in ATP production. The induction of Rheb1 occurs in response to synaptic activity and lactate, and it is actively transported to the mitochondrial matrix through its interaction with Tom20 [67]. However, this overexpression of Rheb2 does not induce any changes in the phosphorylation status of PDH [67].

### 3.3. Rheb1 in Synapse Size and Function

The TSC complex, Rheb1, and mTORC1 are believed to play a crucial role in neuronal and synaptic plasticity through the regulation of protein synthesis [69,111,112,113]. Both synaptic plasticity and memory maintenance rely on mTORC1-mediated translation at synapses and within the cell soma [69,79]. Researchers investigated the impact of TSC complex Rheb1-mTORC1 signaling on synapse construction and function at the Drosophila neuromuscular junction (NMJ) [114,115,116]. While S6K regulated overall synapse growth, Rheb1 overexpression in Drosophila larval motor neurons, or NMJ, led to a doubling in synapse size, specifically the synaptic boutons per muscle region, enhancing synaptic function [117]. Surprisingly, although rapamycin hindered overall development, it did not prevent Rheb1-induced synaptic expansion. The synaptic enlargement induced by Rheb1 was dependent on BMP signaling, mediated by wishful thinking, a BMP type II receptor [115]. This Rheb1-induced synaptic expansion altered the balance between synaptic bouton number and muscle size, affecting normal growth and development [114]. Recent findings suggest that synaptic development influenced by Rheb1 overexpression differed from that of TSC mutants, indicating that the TSC complex might regulate NMJ synapses independently of the Rheb1-TORC1 pathway [116].

In vertebrates, long-term potentiation (LTP) and long-term depression (LTD), and in invertebrates, long-term facilitation (LTF), serve as quantifiable measures of synaptic plasticity [79,118]. Weatherill et al. made a significant discovery on the role of the Rheb1-mTORC1 canonical pathway in memory formation in Aplysia californica, a mollusk. S6K, activated by Rheb1, plays a crucial role in regulating 24h-LTF [119]. Translation mediated by S6K and 4E-BP1 is essential for synaptic plasticity and cognitive processing in both vertebrates and invertebrates [79,119,120]. Loss of phosphatase and tensin homolog (PTEN), a critical negative regulator upstream of TSC complex-Rheb1-mTORC1 signaling, leads to Rheb1 signaling activation, influencing neuronal morphology and social behavior in mice [121]. Rheb1 has been observed to increase acetylcholine and total choline levels in the adult rat brain, both crucial for cognitive function [122].

### 3.4. Rheb1 in Spine Morphology and Function

The TSC complex-Rheb1-mTORC1 pathway is known to influence neuronal architecture and function, and TSC neurological symptoms are, at least in part, owing to changes in cell-autonomous synapse function. Excitatory spine dysmorphogenesis and lower spine synapse density have recently been discovered in TSC neurons of rodent models [86,95]. Independent of mTORC1 activation, Rheb1-syntenin signaling modulates activity-dependent learning and memory processes. Rheb1-GDP binds to syntenin and causes proteasomal breakdown, allowing proper spine production for learning and memory. However, in TSC mutant neurons, Rheb1 remains GTP bound, preventing it from binding to syntenin, resulting in syntenin accumulation and disrupting the link between syndecan-2 and calcium/calmodulin-dependent serine protein kinase. For appropriate spine growth, these proteins must be linked together. As a result, Rheb1 inhibits the establishment of spine synapses in TSC models, resulting in dendritic spine abnormalities [95].

### 3.5. Rheb1 in Axon Guidance

The nervous system’s information flow relies on axons and dendrites. During brain development, neurons establish polarity by growing a single axon and multiple dendrites [123,124]. Components of the TSC complex-Rheb1-mTORC1 pathway have been observed in the growth cones of cortical neuron axons in lab culture [125,126]. These components, along with local protein translation stimulated by signals like Ephrins, support axon growth. They act as upstream regulators, influencing the extension of a single axon through Rap1B GTPases’ control, ensuring the formation of a solitary axon [127,128,129]. According to a study, the overexpression of Rheb1 in the mushroom bodies of Drosophila resulted in larger axonal lobes [130]. While in the visual system, Rheb1 overexpression led to errors in photoreceptor axon guidance but did not affect motor-neuron axon routing, which proceeded normally to synapse at the correct muscle site [114,115]. Proper axon guidance relies on actin regulation, and disruptions in this process might cause molecular defects in the Drosophila visual system [131,132,133]. Moreover, TSC complex-mTORC2 regulation along with Rheb1-mTORC1 growth signaling could influence photoreceptor axon guidance by affecting the actin cytoskeleton. In both Drosophila and mammals, Rheb1-mTORC1 and possibly mTORC2 play roles in establishing neuronal polarity.

### 3.6. Rheb1 in Neuroprotection and Axon Regeneration

Numerous studies have explored Rheb1’s potential to protect neurons by promoting axon regeneration or improving survival in various neuron types [103,134,135]. The idea that the adult mammalian central nervous system (CNS) lacks the ability to regenerate axons after injury has been challenged by recent research. Park et al. demonstrated that continuous activation of Akt-mTORC1 signaling in adult retinal ganglion neurons before optic nerve injury enhanced axon regeneration [102,136]. Akt and Rheb1 play a pivotal role in preserving and regenerating axons not only immediately after injury but also three weeks later, even after the degenerative process has concluded [102,137].

Brain-derived neurotrophic factor (BDNF) and glial cell-derived neurotrophic factor (GDNF) are crucial in safeguarding mature neurons in Parkinson’s disease (PD), marked by the gradual decline of dopaminergic (DA) neurons [102,103,135]. Researchers employed adeno-associated virus constructs expressing a constitutively active form of Rheb1 (AAV-hRheb1 S16H) to shield DA neurons against MPP-induced degeneration in the adult brain. This protection is believed to occur through the elevation of BDNF or GDNF expression and signaling via the mTORC1-CREB pathway [135].

In mouse models of Alzheimer’s disease (AD), the expression of hRheb1 (S16H) in hippocampal neurons boosted the sustained production of BDNF through a mTORC1-dependent pathway, providing neuroprotection in the adult brain [122]. The Akt-Rheb1-mTORC1 signaling pathway plays a role in various aspects of axon growth, such as axon length, count per neuron, and the presence of multiple axons in neurons [125]. When inhibitory molecules were eliminated using chondroitin sulfate-ABC endolyase (ChABC) or combined with Neurotrophin-3, Rheb1 was observed to enhance intrinsic sensory axon regrowth in dorsal root ganglia (DRG) across the dorsal root entry zone (DREZ) [134,138]. Both processes rely on the activation of the Rheb1-mTORC1 pathway, suggesting a shared molecular basis for developmental growth and regrowth following damage [139].

While inhibiting autophagy leads to cell death and neurodegeneration [77,140,141], an acute injury causing excessive autophagy due to the blockade of the Rheb1-mTORC1 pathway might result in axon degeneration [117]. Studies on the regulation of mTOR signaling in neurological and behavioral development indicate that different inputs to mTOR signaling have diverse impacts on nervous system development. However, it is important to consider non-canonical processes in these contexts as well [86,95,114].

## 4. Role of Rheb1 and Rheb2 in Cancer

So far, ten characteristics have been identified as the hallmarks of cancer [142,143]. Moreover, Rheb1 regulates many neuronal activities via mTOR, a protein well-known for being involved in cancer hallmarks. Rheb1’s influence on these hallmarks suggests it could be a potential target for anticancer treatments (Figure 4). Understanding how Rheb1 affects cancer provides insight into its potential role in the pathway leading to tumor formation.

### 4.1. Effects of Rheb1 and Rheb2 on Cancer Hallmarks from Tumor Itself

Previous studies have shown that Rheb1 is highly expressed in various malignant tumors and human malignancies [34,144,145]. Its increased expression has been linked to unfavorable outcomes in several cancer types, including breast cancer, head and neck cancer, skin cancer, colon cancer, prostate cancer, liver cancer, non-small cell lung cancer, and lymphoma [34,145,146,147,148,149,150]. The evidence suggests that Rheb1 plays a role in promoting multiple cancer-causing pathways that could potentially contribute to poorer prognoses in individuals with cancers that exhibit elevated Rheb1 expression [34].

#### 4.1.1. Effect of Rheb1 and Rheb2 on Proliferation

Rheb1 is widely present in various cancer types and contributes to tumor growth by activating mTORC1, a protein often elevated in human cancers like liver, prostate, pancreatic adenocarcinoma, and colorectal cancer [34,145,150,151,152]. This activation leads to increased phosphorylation of p70S6 kinase and 4E-binding protein 1 (4EBP1), enhancing mRNA translation, protein synthesis, and cell proliferation [52,145,153,154,155]. Furthermore, mTORC1 can phosphorylate ULk1 at serine 757, restricting AMPK from phosphorylating and engaging it and hence reducing the autophagic process [52,154,155]. As a result, Rheb1’s enhanced mTORC1 activity hastens the proliferative characteristics of diverse tumor forms [147,148]. In mice, Rheb1 function is required at the early stages of breast cancer progression [156]. Moreover, *Rheb2* was highly expressed in hematopoietic progenitor cell (HSC) populations [157]. Campbell et al. [158] demonstrated that Rheb2 overexpression promoted the development of mouse HSCs. Furthermore, overexpression of Rheb2 positively enhances progenitor cell functions such as colony formation while negatively affecting stem cell competitive repopulation [159]. Although HSCs are not a cancer type, it is suggested that Rheb2 might have a function in promoting the proliferation of cells, including cancer.

Besides activating mTORC1, Rheb1 can impact androgen receptor activity in prostate cancer cells, promoting proliferation [160]. Inhibition of Rheb1 has been shown to slow down the proliferation of colorectal cancer cells and cervical cancer by affecting the mTOR pathway [161,162].

Moreover, Rheb is a member of the Ras family. It is suggested that Rheb might have similar effects on cancer cells as Ras. When it comes to proliferation, KRAS can stimulate cell proliferation in pancreatic cancer [163]. KRAS is in its active GTP-bound state, leading to constant downstream signaling through RAF, PI3K, and other kinases, resulting in uncontrolled cellular proliferation [164]. In addition, HRAS and NF-κB can activate abnormal cell proliferation [165]. NRAS promoted the proliferation of melanocytes [166]. These findings supported the idea that Rheb might be associated with proliferation like Ras. All findings suggest the significant role of Rheb1 in regulating the proliferation of cancer cells.

#### 4.1.2. Effect of Rheb1 on Evading Growth Suppressor

Rheb1 triggers mechanisms that help cancer cells tolerate metabolic stress and survive [167]. For instance, during conditions like serum deprivation, reducing Rheb1 levels through siRNA in the colon cancer cell line Colo320HSR can decrease cell survival [168]. This effect appears to involve Rheb1’s control over p27 protein levels, as reintroducing Rheb1 or p27 can reverse their pro-survival impact [168]. Depletion of both Rheb1 and p27 leads to increased apoptosis [149]. Further studies demonstrated that Rheb1 inhibits apoptosis and promotes the survival of colon cancer cells under serum deprivation by activating autophagy [149]. Additionally, the necessity of Ras activity in repressing p27 levels across the cell cycle has been established [169]. In KRAS mutant colorectal cancer cell lines, a RAS-specific protease has been shown to induce irreversible growth arrest via p27 [170]. These findings indicate that Rheb1 plays a critical role in evading growth suppressors.

#### 4.1.3. Effect of Rheb1 and Rheb2 on Migration, Invasion, and Metastasis

The process of cancer metastasis, where malignant tumors spread to distant areas, often begins with tumor migration [171]. While existing studies primarily highlight Rheb1’s role in metastasis in liver and pancreatic cancers, its significance in this process across various cancers is not well established [146,172]. The functions of Rheb1 have been observed in head and neck cancer, breast cancer, lymphoma, and HCC to enhance invasiveness and metastasis through the mTOR pathway [146,147,148,150]. Moreover, in pancreatic cancer cells, overexpression of Rheb1 accelerated cell migration and invasion, indicating its oncogenic role in this disease [151,152]. Additionally, Rheb2’s involvement in lung and breast cancers suggests its contribution to cancer cell migration, invasion, and metastasis [173]. In addition, mutant K-RAS stimulated invasion and metastasis in pancreatic cancer via GTPase signaling pathways [174]. Oncogenic KRAS is responsible for driving invasion and maintaining metastases in colorectal cancer [175]. These findings highlight the crucial role of Rheb1 and Rheb2 in regulating these processes across various cancers.

#### 4.1.4. Rheb1 on Genomic Instability and Mutation

There is a gap in research concerning the link between Rheb1 and genome instability or mutation. Genome instability is a known factor contributing to mutations in DNA repair genes, often leading to cancer [176]. Studies suggest a connection between mTOR signaling and the DNA damage response (DDR), indicating their interplay in cellular processes [177,178]. Specifically, mTORC1-S6K signaling has been found to potentially increase genome instability by affecting the function of RNF168, an E3 ubiquitin ligase involved in DDR, and this interaction results in the accelerated breakdown of RNF168, impairing its role in maintaining DNA integrity [179]. Given that mTORC1 is downstream of Rheb1, these connections hint at the possibility of Rheb1 playing a role in genome instability and mutation, although specific research on this relationship is currently lacking. RAS contributed to tumorigenesis by causing genomic instability through the imbalanced expression of Aurora-A and BRCA2 urians [180]. It induces chromosome instability and disrupts the DNA damage response [181]. These results indicated the potential involvement of Rheb1 in genomic instability.

#### 4.1.5. Effect of Rheb1 on Resisting Cell Death

Rheb1’s role in cancer is intricate, impacting cell growth and apoptosis, albeit the precise mechanisms remain unclear [160,182]. It influences apoptosis via mTORC1 and independently, particularly through its interaction with FKBP38, Bcl-2, and Bcl-XL [183]. Under optimal conditions, Rheb1 prompts the release of anti-apoptotic proteins from FKBP38, potentially intensifying their interaction with pro-apoptotic proteins and curbing apoptosis.

These anti-apoptotic proteins, Bcl-2 and Bcl-XL, typically migrate to mitochondria through FKBP38 [91]. Rheb1, partly residing in mitochondria, affects FKBP38’s association with these proteins, depending on amino acid and serum levels [184,185]. Elevated Rheb1 activity enhances the Bcl-XL and Bak interaction. When conditions favoring amino acids and serum are met, Rheb1 frees Bcl-2 and Bcl-XL from FKBP38, potentially amplifying their anti-apoptotic influence. This aligns with findings suggesting enhanced Bcl-XL and Bak connections with increased amino acid or Rheb1 expression. Alternatively, this release may prompt FKBP38 to recruit more Bcl-2 and Bcl-XL to mitochondria [91,184].

Rheb1 and FKBP38 seem to mediate interactions between apoptotic and anti-apoptotic proteins. When growth factors are insufficient, Rheb1 inactivation prevents Bcl-2 and Bcl-XL release, potentially restricting their interaction with pro-apoptotic proteins or their presence in mitochondria. This dynamic likely influences cell responses to apoptosis inducers [91].

The mechanism by which Rheb1 regulates FKBP38’s interactions with Bcl-2 and Bcl-XL mirrors its control over FKBP38’s association with mTORC1. Rheb1’s GTP-dependent binding to FKBP38 inhibits its connection with Bcl-2. Upon amino acid or serum stimulation, Rheb1 diminishes these associations, freeing Bcl-2, Bcl-XL, and mTOR from FKBP38 during Rheb1 activation. This pivotal point bifurcates Rheb1’s oncogenic signaling into growth factor control and apoptosis regulation, which is crucial for tumor growth [91].

Calcium and calmodulin modulate FKBP38’s interactions with Bcl-2. They amplify FKBP38’s binding to Bcl-2, countering Rheb1’s effects and shaping FKBP38’s ability to interact with Bcl-2 and Bcl-XL. Rheb1 binds to FKBP38’s FKBP-C domain, which is essential for its interaction with Bcl-2 [100,184,186].

Increased Rheb1 activity is linked to human cancer due to mutations in the TSC1 and TSC2 tumor suppressor genes [187] and aligns with mTORC1’s role in Rheb1’s oncogenic activity. Rheb1’s apoptosis inhibition is crucial in tumorigenesis [147,160].

Rheb1’s hyperactivity could disrupt apoptosis by boosting interactions between pro- and anti-apoptotic proteins, promoting cell survival under conditions inducing apoptosis and leading to uncontrolled growth [91]. Deleting Rheb1 in tubular cells in mice promotes cisplatin-induced tubular cell death [188].

The association between mutated KRAS and various proteins regulating the apoptotic process led to resistance to cell death [189]. KRAS specifically suppressed p53 expression at the protein level during the transformation phase of the blebbishield emergency program, contrasting with apoptotic cells that lack the capability for transformation [190]. HRAS played a role in regulating cell migration and p53-mediated apoptosis from endomembranes [191]. Taken together, these findings indicated that Rheb1 might play a role in resisting cell death.

#### 4.1.6. Effect of Rheb1 on Cancer Metabolism

There is not much proof showing that Rheb1 affects cancer metabolism, but one study suggested that Rheb1-T23M and Rheb1-E40K might impact cell metabolism differently. Rheb1-T23M seems to boost glycolysis rates (how cells convert glucose into energy) more than Rheb1-E40K. Both of these Rheb1 versions enhance cancerous changes in cells through the mTOR signaling pathway [192]. Cells with Rheb1-T23M showed higher levels of a protein called PKM2, indicating they might favor anaerobic glycolysis (a type of energy production) over oxidative phosphorylation [192]. This ‘Warburg effect’ has been linked to cancer cell growth and survival. When these cells were given glucose, those with Rheb1-T23M had much higher glycolysis rates compared to cells with other Rheb1 types [193]. In other words, Rheb1-T23M seems to push cells to rely more on a specific energy production process, likely due to increased PKM2 levels, which could contribute to their increased potential for cancerous changes.

Moreover, Ras mutants are widely recognized for their involvement in regulating glucose metabolism, glutaminolysis, redox homeostasis, lipid metabolism, fatty acid biosynthesis, recycling pathways, and nutrient scavenging [194]. Oncogenic RAS disrupts normal metabolic pathways, leading to the generation of intracellular reactive oxygen species [195]. Given Ras’s significant role in cancer metabolism and the research findings concerning Rheb1 in cancer metabolism, it is suggested that Rheb1 plays a role in reprogramming metabolism in cancer.

#### 4.1.7. Effect of Rheb1 on Enhancing Replicative Immortality

Cellular senescence serves as a crucial failsafe mechanism against transformation and is directly induced by specific oncogenes, including Ras, Akt, and eIF4E [196,197,198]. According to the study, Rheb1 can activate senescence in primary MEFs similarly to Akt or Ras. Rheb1 expression leads to morphological changes, induction of senescence-associated β-galactosidase (SA β-Gal), early growth arrest, and p16 protein induction [147]. As expected, these effects depend on p53 and do not occur in the context of c-Myc expression [147,197]. Importantly, pharmacologic inhibition of mTORC1 with rapamycin can prevent many phenotypic changes associated with senescence induced by Akt and Rheb1 and has a partial effect on Ras-induced senescence. Thus, Rheb1 can block c-Myc-induced apoptosis, and conversely, c-Myc may interfere with senescence activation by Rheb1—likely, these interactions underlie the observed oncogenic cooperation of c-Myc and Rheb1 [147].

Oncogenesis driven by c-Myc is largely opposed to apoptosis [199,200,201], and Rheb1, like eIF4E and Akt, can counter the pro-apoptotic activity of c-Myc. Conversely, c-Myc can interfere with the induction of cellular senescence by Rheb1, Akt, and eIF4E [197]. Acting downstream from mTORC1, eIF4E produces tumors that are phenotypically very similar to Rheb1-driven tumors. However, eIF4E tumors are insensitive to rapamycin. Moreover, eIF4E expression is sufficient to bypass the antitumor effects of rapamycin, indicating that eIF4E is a critical downstream mediator of mTORC1 activity [197]. While Rheb1 certainly has additional targets, it reveals that Rheb1’s transformation-relevant effects on apoptosis, cellular senescence, and chemotherapy responses are strictly dependent on mTORC1 activation and are sensitive to rapamycin. Together, these genetic findings suggest an epistatic relationship between the pro-oncogenic activities of Rheb1, mTORC1, and eIF4E.

In primary pancreatic duct epithelial cells, oncogenic KRAS exerts a suppressive effect on premature senescence. This senescence bypass mechanism involves the upregulation of the basic helix–loop–helix transcription factor Twist, which, in turn, prevents the induction of p16INK4A [202]. KRAS activates the senescence program through three signals [203]: (1) PI3K activation downstream of KRAS [204], (2) Activin A activation [204], and (3) CXCL1 and CXCR2 axes [205]. These findings imply that Rheb1 may play an integral role in cellular immortalization and senescence.

### 4.2. Effects of Rheb1 on Tumor Microenvironments

The tumor microenvironment (TME) has a significant impact, either fostering tumor cell growth or hindering the efficacy of anticancer drugs [206,207]. Therefore, understanding TME and its influencing factors is crucial. TME comprises various cells, like macrophages and fibroblasts. Among these, cancer-associated fibroblasts (CAFs) within the TME play a pivotal role in processes such as proliferation, migration, invasion, and angiogenesis [208,209,210]. We delve into Rheb1’s influence on angiogenesis, tumor-related inflammation, and the suppression of immune responses—key aspects associated with TME and cancer characteristics. Additionally, we explore how Rheb1 affects the connection between cancer and nerves, considering the emerging role of nerve involvement in the TME.

#### 4.2.1. Effect of Rheb1 on Angiogenesis

The process of angiogenesis, the formation of new blood vessels, is controlled by hypoxia-inducible factor (HIF)-mediated transcription of angiogenic factors [211]. A previous study found that Rheb1 overexpression increased HIF1 activity, specifically during low oxygen conditions. Rheb1 heightened HIF1 activity under hypoxia, akin to cells treated with DMOG (a hypoxia-mimicking agent), an effect significantly diminished by rapamycin treatment. Although Rheb1’s increased mTOR activity, as evidenced by heightened Thr389 phosphorylation of S6K1, did not affect the natural levels of HIF1 protein, a similar pattern of HIF1 expression was observed in hypoxic and DMOG-treated cells. This suggests that mTOR seems more focused on regulating HIF1’s transcriptional activity than its stability.

It was demonstrated that Rheb1’s overexpression enhanced HIF1’s transcriptional activity, as evidenced by increased secretion levels of VEGF-A, a well-known gene target of HIF associated with angiogenesis. This highlighted Rheb1’s role in augmenting HIF1 activity within cells. Importantly, the levels of VEGF-A secretion were correlated with HIF activity during hypoxia. Intriguingly, Rheb1 upregulation increased VEGF-A secretion even under rapamycin-sensitive normal oxygen conditions [211].

Ras stimulated angiogenesis by influencing various factors involved in the process, such as vascular endothelial growth factor (VEGF), cyclooxygenases (COX-1 and COX-2), thrombospondins (TSP-1 and TSP-2), urokinase plasminogen activator, and matrix metalloproteases 2 and 9 [212]. HRAS regulates angiogenesis and vascular permeability by activating distinct downstream effectors through Ras, including Raf, p120 Ras GAP, RalGDS, and phosphatidylinositol 3-kinase (PI3K) [213,214]. This suggests that Rheb1 might indeed have a role to play in the process of angiogenesis.

#### 4.2.2. Effect of Rheb1 on Inflammation

Inflammation plays a significant role in tumor development [215,216,217]. Persistent inflammation leads to the production of reactive oxygen and nitrogen species that can damage DNA [218]. Inflammatory responses recruit various cells, including cytokines, chemokines, and enzymes, creating an environment that fosters cancer promotion, progression, and invasion [218,219].

An increased expression of Rheb1 in T-2 toxin-treated RAW264.7 cells notably boosted the production of IL-6, IL-1, and TNF-α, suggesting that Rheb1 overexpression amplifies the expression of inflammatory cytokines triggered by T-2 toxin. Further validation using siRNA confirmed that even when Rheb1 was downregulated, there was still an enhancement in inflammatory cytokine expression, signifying Rheb1 as a promoter of these cytokines [220]. Rheb1, an essential upstream regulator of mTOR, plays a role in inflammation through the Rheb1/mTOR pathway [221,222,223]. Reports show that in H/R-treated HK2 cells, Rheb1 overexpression counteracted the detrimental effects of miR-194 on inflammation (IL-6, IL-1, and TNF-α) [224]. Additionally, Rheb1 siRNA significantly reduced the release of TNF-α, IL17, and IL-23, while transfecting pcDNA3.1(+)-Rheb1 had the opposite effect [225]. Therefore, Rapamycin, a mTOR inhibitor, suppressed the release of the inflammatory cytokines IL-6 and TNF-α [34].

Hence, while the regulation of Rheb1/mTOR in response to inflammation is complex, there are limited data on how the Rheb1/mTOR signaling pathway specifically influences inflammatory cytokines triggered by T-2 toxin. Our investigation revealed that Rheb1 overexpression increased the expression of inflammatory cytokines like IL-6, IL-1, and TNF-α induced by T-2 toxin. Surprisingly, knocking down Rheb1 did not seem to inhibit this response, suggesting other factors might be influencing the expression of these cytokines and warrant further investigation.

Consistent with these findings, the connection between Rheb1 and inflammation appears to involve a feedback control pathway. Higher Rheb1 levels boost inflammatory cytokines, while lower Rheb1 levels reduce inflammation. However, when these cytokines reach a certain low point, their expression rises again. Additionally, studies have identified Rheb1 as a target of miR155-5p, indicating that miR-155-5p can bind to Rheb1’s 3′ UTR and suppress Rheb1 production post-transcriptionally [226]. This aligns with our findings that Rheb1 promotes the expression of inflammatory cytokines triggered by T-2 toxin in RAW264.7 cells [220], suggesting Rheb1’s involvement in modulating inflammation-related elements that facilitate cancer growth.

KRAS mutations are strongly linked to the modulation of tumor inflammation [227]. Oncogenic KRAS triggers the activation of inflammatory cytokines, such as IL-6 and CCL5, during tumorigenesis [228]. Additionally, there is an observed increase in inflammation in aged skin in response to epidermal H-Ras activation [229]. Taken together, it is suggested that Rheb1 might play a role in inflammation.

#### 4.2.3. Effect of Rheb1 on Avoiding Immune Destruction

The effectiveness of anticancer medications is often limited by immune evasion [230]. Interferon regulatory factors (IRFs) are a group of transcription factors that play a crucial role in the immune response against viral infections by regulating interferon-induced immune responses. They also contribute to immune cell development, inflammation, and even oncogenesis [231]. Among the nine known members of the IRF family in mammalian cells (IRF1–IRF9), IRF7, in conjunction with IRF3, is a key regulator of the type 1 interferon response (IFNα/β). IRF7 functions as a transcription factor, activating the transcription of genes stimulated by interferons, including the production of type 1 interferons themselves (IFNα/β). This sets off a positive feedback loop crucial for eliminating viral invasions [232], and it is also associated with antitumor and immunomodulatory roles [233]. The Rheb1/mTOR pathway has been implicated in the activation of IRF7 [234]. Inhibition of the Rheb1/mTOR pathway leads to a decrease in IRF7 levels and dampens the type 1 interferon immune response, potentially affecting the cellular response to IFN-β [235]. Therefore, further research is needed to confirm Rheb1’s impact on the immune system, particularly in the context of cancer.

Mutant KRAS was identified as a key driver of tumor immune evasion in colorectal cancer [236]. Connecting KRAS mutations with tumor-promoting inflammation and immune modulation caused by KRAS leads to immune escape in the TME [237]. These findings suggest that Rheb1 might be associated with immune destruction.

#### 4.2.4. Effect of Rheb1 on Nerve and Cancer Connection

Nerves within the tumor microenvironment significantly impact tumor progression [238,239]. Various cancer types, such as bladder, prostate, pancreatic, colon, lung, head and neck, and bile duct cancers, have been observed to interact with the nervous system, especially in advanced stages [240]. Stress-induced activation of the sympathetic nervous system has been linked to cancer promotion, while reduced tumor nerve connections suggest a better chance of remission [241,242]. β-blockers, known for their anticancer and antiangiogenic actions, enhance chemotherapy responses [243]. The release of neurotransmitters by the hypothalamic–pituitary–adrenal axis (HPA) and the autonomic nervous system (ANS) influences tumor tissue and impacts tumor growth and metastasis [244].

Matrix metalloproteinases (MMPs) play a role in breaking down ECM proteins, cytokine actions, and growth factor synthesis, contributing to the regulation of metastasis by the HPA axis and ANS [245,246].

Previous research indicates that persistent Rheb1 activation in neural progenitor cells (NPCs) affects neuroblast migration [104]. Additionally, Rheb1 boosts the expression of GDNF (glial cell line-derived neurotrophic factor) in the brain by activating mTORC1 [135,247]. GDNF is crucial for neuronal proliferation [248] and has been associated with promoting cancer invasion by positively regulating MMP-9 expression and activity in pancreatic cancer [249]. Although evidence is limited, Rheb1 is believed to influence tumor growth through its involvement in nerve elements. Consequently, Rheb1 appears to play a role in advancing cancer both within cancer cells and in the nerve environment surrounding the tumor, necessitating further comprehensive research.

## 5. Potential Therapeutic Options

To explore therapeutic strategies, we aim to elucidate the interplay among GTPase transducers, including Rho, Ras, and Rheb. This investigation aims to uncover additional avenues for inhibiting Rheb effectors, enhancing our understanding of potential therapeutic interventions.

Concerning the upstream signals involved in the crosstalk between Ras and Rho, it is noteworthy that both Ras and Rho can be jointly regulated by two guanine nucleotide exchange factors (GEFs): Sos and Ras-GRF [250]. Heptahelical receptors can similarly induce the simultaneous activation of Ras and Rho GTPases. Notably, this process is facilitated by another superfamily of GTPases, known as heterotrimeric G proteins [250]. Regarding the downstream signals of Ras and Rho, both Ras and Rho activate PI3K/Akt signaling to promote cell growth. This activation leads to an increase in the expression of genes involved in protein synthesis [251]. While Ras induces p21 through the ERK pathway, Rho inhibits p21 during G1 progression [250]. Indeed, Sos has the ability to stimulate cell migration by activating the Ras–Rho pathway [252]. In the Ras–Rho pathway, Ras can activate Rho through the activation of Phosphoinositide 3-kinases (PI3K) [252].

When it comes to Ras and Rheb, Ras receives signals from proteins like Sos or Ras-GRF. However, Rheb does not necessarily require a dedicated GEF since its basal nucleotide exchange rates are sufficiently high to load it with GTP effectively [65]. Regarding the downstream effects of Ras and Rheb, Ras can hinder the TSC1/TSC2 complex through PI3K activation. This inhibition of the TSC complex by Ras-GTP enables Rheb to activate mTOR, the master regulator of translation [253,254]. In addition to the effect of Ras on Rheb activation, Rheb is activated by NOX2. Upon activation, NOX2 produces superoxide, which is then converted to hydrogen peroxide, likely by superoxide dismutase. Hydrogen peroxide inhibits TSC1/2, and when TSC1/2 is suppressed, it increases the levels of RheB-GTP. This, in turn, enhances the activity of mTORC1 [255].

As of now, there are no reports on the crosstalk between Rho and Rheb. However, it is worth noting that both Rheb and Rho could potentially act as downstream effectors of Ras and be regulated by Ras through the Akt-PI3K pathway. It is suggested that Ras inhibitors might be indirect Rheb inhibitors, such as farnesyltransferase inhibitors.

Currently, there appear to be limited studies on Rheb1 inhibitors. Reports have categorized Rheb1 inhibitors into two main groups: farnesyltransferase inhibitors (FTIs) [256,257] and direct binders [258] (Figure 5).

### 5.1. Direct Binders: NR1 Inhibitors

NR1 is a small molecule that directly binds to Rheb1, showing promise in cancer prevention. It targets the switch II domain of Rheb1, specifically inhibiting mTORC1 activity. Using isothermal calorimetric (ITC) techniques, NR1 binding to Rheb1 was quantified despite its low aqueous solubility. It was found to bind Rheb1 with a KD = 1.5 M and a 1:1 stoichiometry. The binding energy of NR1 was mainly driven by entropy (H = −0.315 kcal/mol; TS = −0.7628 kcal/mol), indicating hydrophobic interactions as the primary cause of NR1 binding to Rheb1. This aligns with the intermolecular contacts observed in the X-ray crystal structure [258].

Several lines of evidence suggest that NR1 selectively interacts with Rheb1 in cells with single-digit micromolar potencies. Firstly, NR1 induces a signaling pattern similar to that observed in Rheb1 knockout embryos and MEFs [259,260]. NR1 treatment suppresses mTORC1 kinase, the primary target of Rheb1, while elevating mTORC2 kinase activity. This relief of an mTORC1-dependent negative feedback loop activates mTORC2, a complex sharing the same mTOR catalytic subunit as mTORC1. Secondly, NR1 does not directly inhibit mTOR kinase in an in vitro kinase assay and does not inhibit mTORC2 in cells, indicating that NR1 does not directly target mTOR kinase. Instead, NR1 can still suppress mTORC1 signaling in TRI102 cells, suggesting its target lies between TSC and mTORC1. Lastly, NR1 does not diminish Ras signaling in cells or Rap1 function in vitro, despite their similarities to small G-proteins like Rheb1.

Recent research has identified NR1 as a direct binder of Rheb1, demonstrating the effectiveness of pharmacologically targeting Rheb1. Inhibiting Rheb1 directly leads to a distinct signaling pattern compared to current mTOR pathway inhibitors, particularly in terms of mTORC2 inhibition, potentially offering different clinical benefits [258].

### 5.2. Farnesyltransferase Inhibitors

Lonafarnib, Tipifarnib, and FTI-277 belong to the class of FTIs, inhibitors of Rheb1 [256,257]. Lonafarnib, specifically, is a nonpeptidic CaaX-competitive selective inhibitor of FTase (FTase IC50 = 1.9 nM) [261,262]. It effectively suppresses Rheb1 prenylation, demonstrating its ability to hinder alternative prenylation pathways. Lonafarnib treatment results in reduced phosphorylation of S6 ribosomal protein, a downstream target of Rheb1 and mTOR signaling [256]. Moreover, lonafarnib’s ability to enhance tamoxifen and taxane-induced apoptosis is nullified by a mutant geranylgeranylated Rheb1 protein, Rheb1-CSVL, which has been derived from Rheb1-CSVM. This underscores lonafarnib’s role in replicating the effects of dominant negative Rheb1 or Rheb1 short interfering RNA (siRNA) in sensitizing tumor cells to apoptosis induced by tamoxifen and taxanes [256]. Noticeably, lonafarnib at a concentration of 5 μM did not inhibit Ras-GTP and its downstream effectors, such as MAPK and Akt [263].

Tipifarnib, identified in a multi-institution phase II study, is an orally bioavailable, non-peptidomimetic FTI [264]. It inhibits FTase prenylation of K-Ras in vitro with an IC50 of 7.9 nM and shows high selectivity against GGTase-1 (40% inhibition at 50,000 nM) [265]. Moreover, Tipifarnib significantly inhibited HRAS farnesylation in head and neck squamous cell carcinomas [266] and HRAS in rhabdomyosarcoma [267]. Tipifarnib reduces the mobility of Rheb1 and inhibits its prenylation, along with suppressing mTOR phosphorylation on Ser2448, p70S6 kinase, and ribosomal S6, without affecting the phosphorylation of the Rheb1 regulator TSC2 or other Akt substrates in the early stages [257].

Limited research has been conducted on FTI-277 inhibitors. However, it has been observed that FTI-277 inhibits Rheb1-induced phosphorylation of S6K [16,268]. These findings suggest that FTIs comprehensively suppress the Rheb1 signaling pathway more extensively than mTOR/Raptor inhibitors like rapamycin. FTIs effectively block Rheb1-dependent signaling, encompassing the regulation of mTOR/Raptor governing growth and mTOR/Rictor controlling the actin cytoskeleton. Additionally, FTI-277 inhibited breast cell invasion and migration by blocking HRAS activation [269].

FTIs might inhibit the post-translational lipid modification of HRAS. While KRAS and NRAS are also substrates of farnesyl transferase, FTIs failed to prevent the prenylation of K- and N-Ras. Compellingly, FTIs still inhibited the growth of mouse mammary tumor virus (MMTV)-KRAS and MMTV-NRAS tumors in preclinical models [256]. Additionally, research on the moslecular biology of the defective pathway has primarily concentrated on the activation of H-Ras. However, the activation of K-Ras or other farnesylated proteins may be more critical in tumorigenesis [270]. This suggests that the anti-proliferative action of FTIs is dependent on blocking the farnesylation of other proteins, such as Rheb [256].

In a recent study, although Simvastatin is an HMG-CoA reductase inhibitor, it can impair the prenylation of Rheb in U937/WT, U937/CHR2863R0.2, and U937/CHR2863R5 cells [271].

## 6. Future Perspectives

In this review, we provided a concise overview of Rheb1 as both a biomarker and a novel target in cancer development. Rheb1 tends to be overexpressed in various cancers, and several studies link its upregulation with cancer development. While there is extensive research on Rheb1’s impact on cancer hallmarks, some aspects remain unstudied and require further investigation for confirmation.

Most of Rheb1’s functions operate through the Rheb1/mTORC1 signaling pathway. The primary driver of Rheb1’s cancer-causing activity is its ability to activate the mTORC1 pathway. Yet, the potential mTORC1-independent processes induced by Rheb1 in carcinogenesis remain unclear. These may contribute to resistance against mTORC1 inhibitors, prompting a need for a comprehensive understanding of Rheb1 signaling in cancer therapy to design more effective treatment strategies.

FTIs inhibit Rheb1 prenylation, potentially contributing to their anti-proliferative effects. However, their lack of selectivity across various GTPases limits their effectiveness as specialized Rheb1-targeted drugs. Recent studies highlighting NR1 as a small molecule directly binding Rheb1 and showing potential in cancer prevention offer promising insights. Moreover, a recent study showed Simvastatin, an HMG-CoA reductase inhibitor, suppresses prenylation of Rheb. It is suggested that other HMG-CoA reductase inhibitors, such as Lovastatin and Pravastatin, might inhibit the prenylation of Rheb. In addition, NOX2 inhibitors might be a potential option for blocking Rheb1-GTP via quenching reactive oxygen species (H_2_O_2_). However, further research is necessary to identify specific inhibitors targeting Rheb1 that can preferentially inhibit cancer growth.

Applying researchers’ techniques across multiple Rheb1-related carcinomas could potentially reconcile the conflicting findings associated with Rheb1 in different cancers. Clarifying Rheb1’s role in cancer development and progression could pave the way for the accelerated development of Rheb1-specific inhibitors, offering a new approach to combating refractory cancers in the future.

## Figures and Tables

**Figure 1 ijms-25-01489-f001:**
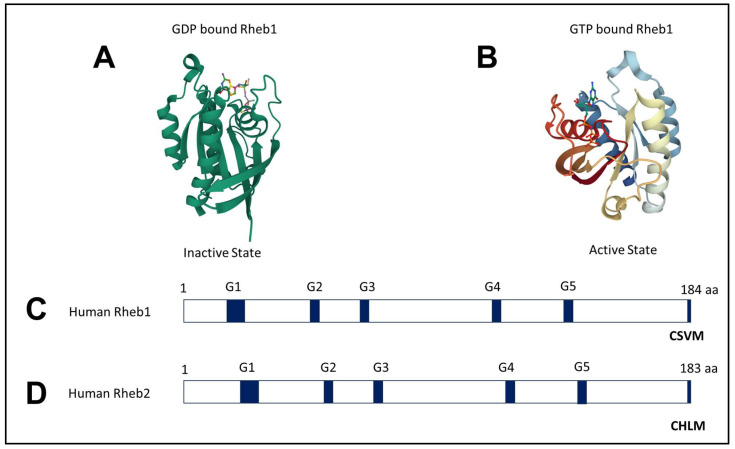
Crystal Structure of Rheb1: (**A**) The crystal structures of Rheb1 bound with GDP (Inactive State) and (**B**) Rheb1 bound with GTP (Active State) were aligned. (**C**) Rheb1 protein, with a length of 184 amino acids, is represented with the approximate locations of the G-boxes 1–5 and CSVM (CAAX motif) [23]. (**D**) Rheb2 protein, with a length of 183 amino acids, is represented by the approximate locations of the G-boxes 1–5 and CHLM (CAAX motif) [23]. Crystal structures are from the Protein Data Bank (PDB ID: 1XTQ, 1XTS).

**Figure 2 ijms-25-01489-f002:**
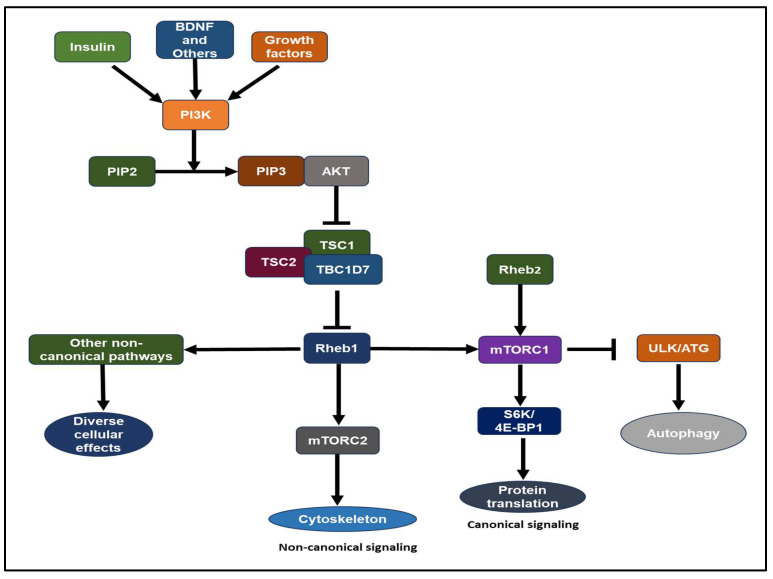
The Rheb1 and Rheb2-mTOR signaling pathways have been extensively studied, from yeast to humans. This pathway plays a pivotal role in the regulation of cell growth, neurons, proliferation, cell size, and cancer proliferation. Canonical pathway: translation and cell cycle progression via S6K and 4E-BP1 inhibit autophagy via Ulk and ATG proteins. Non-canonical: mTORC2 regulates the cytoskeleton. Abbreviations: G-protein-coupled receptors (GPCRs), Insulin-like growth factor 1 (IGF-1), Mechanistic Target of Rapamycin (mTOR), Phosphatidylinositol-3 kinase (PI3K), Phosphatidylinositol 4,5-bisphosphate (PIP2), Phosphatidylinositol 3,4,5-trisphosphate (PIP3), Rheb1 GTPase, TBC1D7, Tuberous Sclerosis Complex (TSC) complex.

**Figure 3 ijms-25-01489-f003:**
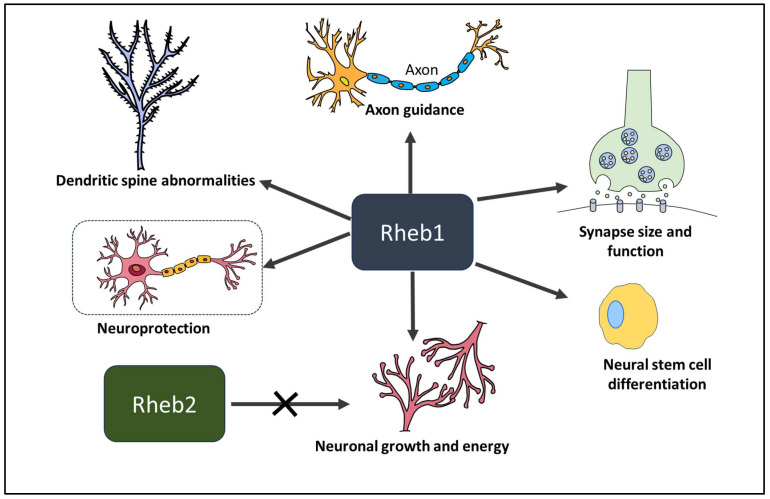
An overview of the different biological activities mediated by Rheb1. Rheb1 is crucial for numerous biological processes in addition to the central nervous system. Function in the central nervous system includes neural stem cell differentiation, neuronal growth and energy, synapse function, axon guidance, and neuroprotection.

**Figure 4 ijms-25-01489-f004:**
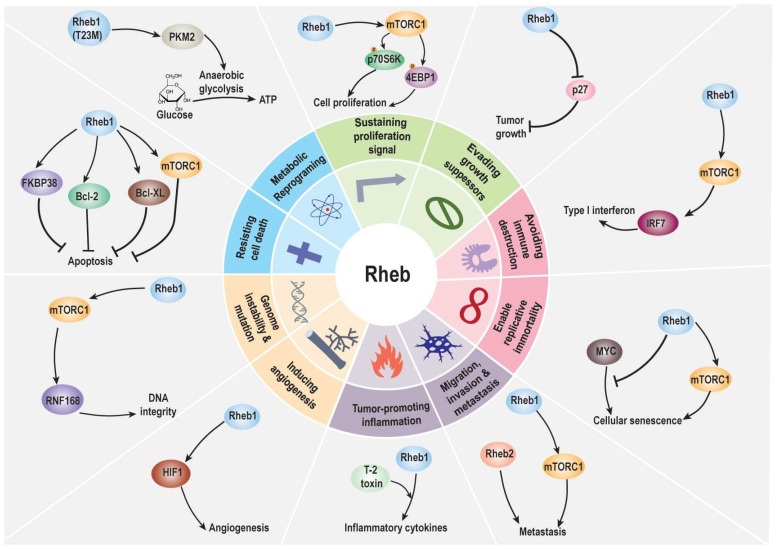
Hallmarks of cancer include sustained proliferation signals, cancer cell survival, migration, invasion, and metastasis, resisting cell death, angiogenesis, possible replication immortality, inflammation promotion of cancer, avoidance of immune destruction, genetic instability and mutations, and metabolic reprogramming. Seven of the cancer hallmarks originate from tumor cells themselves, while the rest are involved in the tumor microenvironment (TME). Abbreviations: Bcl-2: B-cell lymphoma 2; Bcl-XL: B-cell lymphoma-extra-large; FKBP38: FK506-binding protein 38; HIF1: Hypoxia-inducible factor 1; IRF7: Interferon regulatory factor 7; MYC: Myelocytomatosis oncogene; mTORC1: Mechanistic target of rapamycin complex 1; p27: Cyclin-dependent kinase inhibitor 1B (CDKN1B); p70S6 kinase: S6 kinase 1 (RPS6KB1); PKM2: Pyruvate kinase isozyme M2; RNF168: Ring finger protein 168; Rheb1: Ras homolog enriched in brain 1; 4EBP1: Eukaryotic translation initiation factor 4E-binding protein 1.

**Figure 5 ijms-25-01489-f005:**
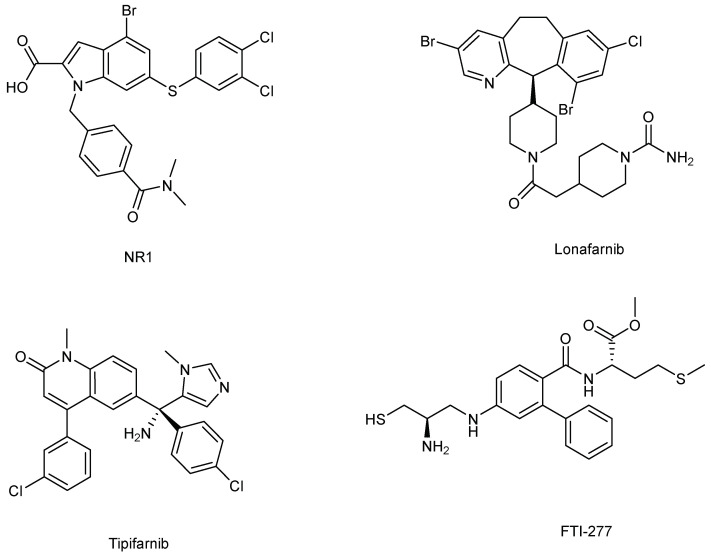
The structures of Rheb1 inhibitors: NR1, Lonafarnib, Tipifarnib, FTI-277.

**Table 1 ijms-25-01489-t001:** Information about mutation types in Rheb1. Asterisk (*) denotes a stop codon.

Mutation Type	Amino Acid Position	Mutation (s)	References
Activating	16	S16H	[42]
	35	Y35N/C/H	[43]
	63	G63A	[44]
	64	Q64L	[45]
	153	N153T/S	[42,46]
Loss of function	35	Y35A	[30,46]
	36	D36A	[21]
	37	P37A	[21,46]
	38	T38A	[21,46,47]
	39	I39A, K	[46,47]
	41	N41A	[21,47]
	65	D65A	[30]
	67,69	Y67A/I69A	[48]
	76,77	I76A/D77A	[48]
Activating Mutations Found in Cancer Database	35	Y35N (5×)	[43,49]
	139	E139K/D/G/* (2×)	[43]
Loss of Membrane Association	181	C181S	[20,33]
Dominant Negative	20	S20N	[12,21]
	60	D60I/K/V	[12,21]

## Data Availability

Not applicable.

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
