# Peer review of "Unraveling the Role of Ras Homolog Enriched in Brain (Rheb1 and Rheb2): Bridging Neuronal Dynamics and Cancer Pathogenesis through Mechanistic Target of Rapamycin Signaling"

_ijms, 2024, doi:10.3390/ijms25031489_

Round 1

Reviewer 1 Report

Comments and Suggestions for Authors

The current review article explores the multifaceted roles of Rheb1 and Rheb2 in cellular processes and in cancer, and their potential as targets for anti-cancer therapies. The review is interesting and well-written, however some issues need to be addressed:

1. The manuscript reviews also the role of Rheb in Neurons, the emphasis is not only on cancer, please modify the Abstract and Title respectively.

2. Figure 4 is not convincing to be part of the review, Figure 4 contains the same as previous publications, it does not present anything more, please modify the Figure to add more specific the role of Rheb in each case. 

3. There is a lack of recent publications in the field, please update references and add recent literature in the field.

Comments on the Quality of English Language

Minor editing of English language is required.

Author Response

Comments and Suggestions for Authors

The current review article explores the multifaceted roles of Rheb1 and Rheb2 in cellular processes and in cancer, and their potential as targets for anti-cancer therapies. The review is interesting and well-written, however some issues need to be addressed:

  1. The manuscript reviews also the role of Rheb in Neurons, the emphasis is not only on cancer, please modify the Abstract and Title respectively.

Response) Thank you for your comments. We modify the abstract and title as follows.

Title: Unraveling the Role of Rheb1 and Rheb2: Bridging Neuronal Dynamics and Cancer Pathogenesis through mTOR Signaling

Abstract: Ras homolog enriched in brain (Rheb1 and Rheb2), small GTPases, play a crucial role in regulating neuronal activity and have gained attention for their implications in cancer development, particularly in breast cancer. This study delves into the intricate connection between the multifaceted functions of Rheb1 in neurons and cancer, with a specific focus on the mTOR pathway. It aims to elucidate Rheb1's involvement in pivotal cellular processes such as proliferation, apoptosis resistance, migration, invasion, metastasis, and inflammatory responses, while acknowledging that Rheb2 has not been extensively studied. Despite the recognized associations, a comprehensive understanding of the intricate interplay between Rheb1, Rheb2, and their roles in both nerve and cancer remains elusive.

This review consolidates current knowledge regarding the impact of Rheb1 on cancer hallmarks and explores the potential of Rheb1 as a therapeutic target in cancer treatment. It emphasizes the necessity for a deeper comprehension of the molecular mechanisms underlying Rheb1-mediated oncogenic processes, underscoring the existing gaps in our understanding. Additionally, the review highlights the exploration of Rheb1 inhibitors as a promising avenue for cancer therapy.

By shedding light on the complicated roles between Rheb1/Rheb2 and cancer, this study provides valuable insights to the scientific community. These insights are instrumental in guiding the identification of novel targets and advancing the development of effective therapeutic strategies for treating cancer.

  1. Figure 4 is not convincing to be part of the review, Figure 4 contains the same as previous publications, it does not present anything more, please modify the Figure to add more specific the role of Rheb in each case.

Response) Thank you for your comment. We modified Figure 4 and added more specific information about the role of Rheb in this figure (Location: line 451-461).

  1. There is a lack of recent publications in the field, please update references and add recent literature in the field.

Response) Thank you for your comment. We added more recent publication in the field as followings:

In KRAS mutant colorectal cancer cell lines, a RAS-specific protease has been shown to induce irreversible growth arrest via p27 [170].

  1. Stubbs, C.K.; Biancucci, M.; Vidimar, V.; Satchell, K.J.F. RAS specific protease induces irreversible growth arrest via p27 in several KRAS mutant colorectal cancer cell lines. Scientific Reports 2021, 11, 17925. https://doi:10.1038/s41598-021-97422-0.

In addition, mutant K-RAS stimulated invasion and metastasis in pancreatic cancer via GTPase signaling pathways [174]. Oncogenic KRas was responsible for driving invasion and maintaining metastases in colorectal cancer [175].

  1. Boutin, A.T.; Liao, W.T.; Wang, M.; Hwang, S.S.; Karpinets, T.V.; Cheung, H.; Chu, G.C.; Jiang, S.; Hu, J.; Chang, K.; et al. Oncogenic Kras drives invasion and maintains metastases in colorectal cancer. Genes Dev 2017, 31, 370-382. https://doi:10.1101/gad.293449.116.

The association between mutated KRas and various proteins regulating the apoptotic process led to resistance to cell death [189]. KRas specifically suppressed p53 expression at the protein level during the transformation phase of the blebbishield emergency program, contrasting with apoptotic cells that lack the capability for transformation [190]. HRAS played a role in regulating cell migration and p53-mediated apoptosis from endomembranes [191].

  1. Ferreira, A.; Pereira, F.; Reis, C.; Oliveira, M.J.; Sousa, M.J.; Preto, A. Crucial Role of Oncogenic KRAS Mutations in Apoptosis and Autophagy Regulation: Therapeutic Implications. Cells 2022, 11, https://doi:10.3390/cells11142183.
  2. Godwin, I.; Anto, N.P.; Bava, S.V.; Babu, M.S.; Jinesh, G.G. Targeting K-Ras and apoptosis-driven cellular transformation in cancer. Cell Death Discovery 2021, 7, 80. https://doi:10.1038/s41420-021-00457-5.
  3. Santra, T.; Herrero, A.; Rodriguez, J.; von Kriegsheim, A.; Iglesias-Martinez, L.F.; Schwarzl, T.; Higgins, D.; Aye, T.T.; Heck, A.J.R.; Calvo, F.; et al. An Integrated Global Analysis of Compartmentalized HRAS Signaling. Cell Rep 2019, 26, 3100-3115.e3107. https://doi:10.1016/j.celrep.2019.02.038.

Moreover, Ras mutants are widely recognized for their involvement in regulating glucose metabolism, glutaminolysis, redox homeostasis, lipid metabolism, fatty acid bio-synthesis, recycling pathways, and nutrient scavenging [194]. Oncogenic RAS disrupts normal metabolic pathways, leading to the generation of intracellular reactive oxygen spe-cies [195].

  1. Mukhopadhyay, S.; Vander Heiden, M.G.; McCormick, F. The Metabolic Landscape of RAS-Driven Cancers from biology to therapy. Nat Cancer 2021, 2, 271-283. https://doi:10.1038/s43018-021-00184-x.
  2. Bartolacci, C.; Andreani, C.; El-Gammal, Y.; Scaglioni, P.P. Lipid Metabolism Regulates Oxidative Stress and Ferroptosis in RAS-Driven Cancers: A Perspective on Cancer Progression and Therapy. Front Mol Biosci 2021, 8, 706650. https://doi:10.3389/fmolb.2021.706650

KRas activates the senescence program through three signals [203]: (1) PI3K activation, downstream of Kras [204], (2) Activin A activation [204], (3) CXCL1 and CXCR2 axis [205].

  1. Yang, K.; Li, X.; Xie, K. Senescence program and its reprogramming in pancreatic premalignancy. Cell Death Dis 2023, 14, 528. https://doi:10.1038/s41419-023-06040-3.
  2. Zhao, Y.; Wu, Z.; Chanal, M.; Guillaumond, F.; Goehrig, D.; Bachy, S.; Principe, M.; Ziverec, A.; Flaman, J.-M.; Collin, G. Oncogene-induced senescence limits the progression of pancreatic neoplasia through production of activin A. Cancer Research 2020, 80, 3359-3371.

KRAS mutations are strongly linked to the modulation of tumor inflammation [227]. Oncogenic KRAS triggers the activation of inflammatory cytokines, such as IL-6 and CCL5, during tumorigenesis [228]. Additionally, there is an observed increase in inflammation in aged skin in response to epidermal H-Ras activation [229]. Taken together, it is suggested that Rheb1 might play a role in inflammation.

  1. Pereira, F.; Ferreira, A.; Reis, C.A.; Sousa, M.J.; Oliveira, M.J.; Preto, A. KRAS as a Modulator of the Inflammatory Tumor Microenvironment: Therapeutic Implications. Cells 2022, 11, https://doi:10.3390/cells11030398.

Mutant KRAS was identified as a key driver of tumor immune evasion in colorectal cancer [236]. Connecting KRAS mutations with tumour-promoting inflammation and immune modulation caused by KRAS that leads to immune escape in the TME [237]. These findings suggest that Rheb1 might be associated with immune destruction.

  1. Liu, H.; Liang, Z.; Cheng, S.; Huang, L.; Li, W.; Zhou, C.; Zheng, X.; Li, S.; Zeng, Z.; Kang, L. Mutant KRAS Drives Immune Evasion by Sensitizing Cytotoxic T-Cells to Activation-Induced Cell Death in Colorectal Cancer. Adv Sci (Weinh) 2023, 10, e2203757. https://doi:10.1002/advs.202203757.
  2. Hamarsheh, S.; Groß, O.; Brummer, T.; Zeiser, R. Immune modulatory effects of oncogenic KRAS in cancer. Nat Commun 2020, 11, 5439. https://doi:10.1038/s41467-020-19288-6

In Ras-Rho pathway, Ras can activate Rho through the activation of Phosphoinosi-tide 3-kinases (PI3K) [252].

When it comes to Ras and Rheb, Ras receives signals from proteins like Sos or Ras-GRF. However, Rheb doesn't necessarily require a dedicated GEF since its basal nu-cleotide exchange rates are sufficiently high to load it with GTP effectively [253].

  1. Soriano, O.; Alcón-Pérez, M.; Vicente-Manzanares, M.; Castellano, E. The Crossroads between RAS and RHO Signaling Pathways in Cellular Transformation, Motility and Contraction. Genes (Basel) 2021, 12, https://doi:10.3390/genes12060819.
  2. Schöpel, M.; Potheraveedu, V.N.; Al-Harthy, T.; Abdel-Jalil, R.; Heumann, R.; Stoll, R. The small GTPases Ras and Rheb studied by multidimensional NMR spectroscopy: structure and function. Biol Chem 2017, 398, 577-588. https://doi:10.1515/hsz-2016-0276.

Upon activation, NOX2 produces superoxide, which is then converted to hydrogen peroxide, likely by superoxide dismutase. Hydrogen peroxide inhibits TSC1/2, and when TSC1/2 is suppressed, it increases the levels of RheB-GTP. This, in turn, enhances the activity of mTORC1 [256].

  1. Chen, J.; Liu, C.; Chernatynskaya, A.V.; Newby, B.; Brusko, T.M.; Xu, Y.; Barra, J.M.; Morgan, N.; Santarlas, C.; Reeves, W.H.; et al. NADPH Oxidase 2-Derived Reactive Oxygen Species Promote CD8+ T Cell Effector Function. J Immunol 2024, 212, 258-270. https://doi:10.4049/jimmunol.2200691.

In a recent study, although Simvastatin is HMG-CoA reductase inhibitors, Simvas-tatin can impair the prenylation of Rheb in U937/WT, U937/CHR2863R0.2 and U937/CHR2863R5 cells [273].

  1. Jansen, G.; Al, M.; Assaraf, Y.G.; Kammerer, S.; van Meerloo, J.; Ossenkoppele, G.J.; Cloos, J.; Peters, G.J. Statins markedly potentiate aminopeptidase inhibitor activity against (drug-resistant) human acute myeloid leukemia cells. Cancer Drug Resist 2023, 6, 430-446. https://doi:10.20517/cdr.2023.20

Reviewer 2 Report

Comments and Suggestions for Authors

This review analyse in detail the role of Rheb 1 and in part of Rheb 2 on neurons and cancer cells.

Actually, from the title, the review should be focused on cancer cells. However, this topic is considered starting from the page 12.

Of course, it is necessary a detailed analysis of the functions of Rheb protein.

I would say that the review is of interest and deals with an interesting topic.

I am not an expert on this specific GTPase, but actually I know that several other GTPase, some with strong similarities, can be involved in several of the biological processes where Rheb is involved. This is certainly true in the case of cancer.

I would shorten the part of the review regarding the neurons, or alternatively I would change the title, inserting this topic. Otherwise, some of the findings reported are not related to cancer.

It should be of interest to understand and depict the possible relationship among Rheb and Ras and Rho. Indeed, there should be a relationship among these signal transducers, and it is clear (or it is possible) that the therapeutic approaches suggested for blocking Rheb could affect the other transducers mentioned.

Actually, FTI (and GTI) inhibitors can have different target molecules other than Rheb.

In other words, the authors should take into account how much the function of a given transducer is dependent on or independent of the others.  Also, the possibility that one transducer can substitute the other. This should be considered in detail with therapeutic options proposed.

Some references, in which Rheb is mentioned, should be considered and discussed. Alternatively, the author should explain why they do not take them into account.

NADPH Oxidase 2-Derived Reactive Oxygen Species Promote CD8+ T Cell Effector Function. Chen J, Liu C, Chernatynskaya AV, Newby B, Brusko TM, Xu Y, Barra JM, Morgan N, Santarlas C, Reeves WH, Tse HM, Leiding JW, Mathews CE.J Immunol. 2023 Dec 11;212(2):258-70. doi: 10.4049/jimmunol.2200691. Online ahead of print.PMID: 38079221    

Statins markedly potentiate aminopeptidase inhibitor activity against (drug-resistant) human acute myeloid leukemia cells.

Jansen G, Al M, Assaraf YG, Kammerer S, van Meerloo J, Ossenkoppele GJ, Cloos J, Peters GJ. Cancer Drug Resist. 2023 Jul 4;6(3):430-446. doi: 10.20517/cdr.2023.20. eCollection 2023.PMID: 37842233

Comments on the Quality of English Language

English language is good.

Author Response

Comments and Suggestions for Authors

This review analyse in detail the role of Rheb 1 and in part of Rheb 2 on neurons and cancer cells.

Actually, from the title, the review should be focused on cancer cells. However, this topic is considered starting from the page 12.

Of course, it is necessary a detailed analysis of the functions of Rheb protein.

  1. I would say that the review is of interest and deals with an interesting topic. I am not an expert on this specific GTPase, but actually I know that several other GTPase, some with strong similarities, can be involved in several of the biological processes where Rheb is involved. This is certainly true in the case of cancer.

Response) Thank you for your great comment, we added more information about KRas, HRas, NRas in the hallmarks of cancer as followings

Location: line 483-491:

Moreover, Rheb is a member of the Ras family. It is suggested that Rheb might have similar effects on cancer cell like Ras. When it comes to proliferation, KRas can stimulate cell proliferation in pancreatic cancer [163]. KRAS in its active GTP-bound state, leading to constant downstream signaling through RAF, PI3K, and other kinases, resulting in uncontrolled cellular proliferation [164]. In addition, HRas and NF-κB can activate abnormal cell proliferation [165]. NRas promoted the proliferation of melanocytes [166] these findings supported that Rheb might be associated with proliferation like Ras. All findings suggest the significant role of Rheb1 in regulating the proliferation of cancer cells.

Location: line 500-504:

Additionally, the necessity of Ras activity in repressing p27 levels across the cell cy-cle has been established [169]. In KRAS mutant colorectal cancer cell lines, a RAS-specific protease has been shown to induce irreversible growth arrest via p27 [170]. These findings indicate that Rheb1 plays a critical role in evading growth suppressors.

Location: line 515-519:

In addition, mutant K-RAS stimulated invasion and metastasis in pancreatic cancer via GTPase signaling pathways [174]. Oncogenic KRas was responsible for driving inva-sion and maintaining metastases in colorectal cancer [175]. These findings highlight the crucial role of Rheb1 and Rheb2 in regulating these processes across various cancers.

Location: 532-535:

RAS contributed to tumorigenesis by causing genomic instability through the imbal-anced expression of Aurora-A and BRCA2 urian [180]. It induced chromosome instability and disrupts the DNA damage response [181]. These results indicated a potential in-volvement of Rheb1 in genomic instability

Location: line 574-580:

The association between mutated KRas and various proteins regulating the apoptotic process led to resistance to cell death [189]. KRas specifically suppressed p53 expression at the protein level during the transformation phase of the blebbishield emergency program, contrasting with apoptotic cells that lack the capability for transformation [190]. HRAS played a role in regulating cell migration and p53-mediated apoptosis from endomembranes  [191]. Taken together, these given findings indicated that Rheb1 might play a role in resisting cell death.

Location: line 594-600:

Moreover, Ras mutants are widely recognized for their involvement in regulating glucose metabolism, glutaminolysis, redox homeostasis, lipid metabolism, fatty acid bio-synthesis, recycling pathways, and nutrient scavenging [194]. Oncogenic RAS disrupts normal metabolic pathways, leading to the generation of intracellular reactive oxygen spe-cies [195]. Given Ras's significant role in cancer metabolism and the research findings concerning Rheb1 in cancer metabolism, it is suggested that Rheb1 plays a role in repro-gramming metabolism in cancer.

Location: line 626-632:

In primary pancreatic duct epithelial cells, oncogenic KRas exerts a suppressive effect on premature senescence. This senescence bypass mechanism involves the upregulation of the basic helix-loop-helix transcription factor Twist, which, in turn, prevents the induction of p16INK4A [202]. KRas activates the senescence program through three signals [203]: (1) PI3K activation, downstream of Kras [204], (2) Activin A activation [204], (3) CXCL1 and CXCR2 axis [205]. These findings imply that Rheb1 may play an integral role in cellular immortalization and senescence.

Location: line 661-667:

Ras stimulated angiogenesis by influencing various factors involved in the process, such as vascular endothelial growth factor (VEGF), cyclooxygenases (COX-1 and COX-2), thrombospondins (TSP-1 and TSP-2), urokinase plasminogen activator and matrix metalloproteases-2 and -9 [212]. HRas regulated angiogenesis and vascular permeability by activation of distinct downstream effectors through Ras, including Raf, p120 Ras GAP, RalGDS, phosphatidylinositol 3-kinase (PI3K) [213,214]. This suggests that Rheb1 might indeed have a role to play in the process of angiogenesis

Location: line 702-706

KRAS mutations are strongly linked to the modulation of tumor inflammation [227]. Oncogenic KRAS triggers the activation of inflammatory cytokines, such as IL-6 and CCL5, during tumorigenesis [228]. Additionally, there is an observed increase in inflammation in aged skin in response to epidermal H-Ras activation [229]. Taken together, it is suggested that Rheb1 might play a role in inflammation.

Location: line 723-726

Mutant KRAS was identified as a key driver of tumor immune evasion in colorectal cancer [236]. Connecting KRAS mutations with tumour-promoting inflammation and immune modulation caused by KRAS that leads to immune escape in the TME [237]. These findings suggest that Rheb1 might be associated with immune destruction.

Q2. I would shorten the part of the review regarding the neurons, or alternatively I would change the title, inserting this topic. Otherwise, some of the findings reported are not related to cancer.

Response) Thank you for your great comment, we modified as below

Title: Unraveling the Role of Rheb1 and Rheb2: Bridging Neuronal Dynamics and Cancer Pathogenesis through mTOR Signaling

Abstract: : Ras homolog enriched in brain (Rheb1 and Rheb2), small GTPases, play a crucial role in regu-lating neuronal activity and have gained attention for their implications in cancer development, particularly in breast cancer. This study delves into the intricate connection between the multi-faceted functions of Rheb1 in neurons and cancer, with a specific focus on the mTOR pathway. It aims to elucidate Rheb1's involvement in pivotal cellular processes such as proliferation, apop-tosis resistance, migration, invasion, metastasis, and inflammatory responses, while acknowl-edging that Rheb2 has not been extensively studied. Despite the recognized associations, a com-prehensive understanding of the intricate interplay between Rheb1, Rheb2, and their roles in both nerve and cancer remains elusive. This review consolidates current knowledge regarding the impact of Rheb1 on cancer hallmarks and explores the potential of Rheb1 as a therapeutic target in cancer treatment. It emphasizes the necessity for a deeper comprehension of the molec-ular mechanisms underlying Rheb1-mediated oncogenic processes, underscoring the existing gaps in our understanding. Additionally, the review highlights the exploration of Rheb1 inhibi-tors as a promising avenue for cancer therapy. By shedding light on the complicated roles be-tween Rheb1/Rheb2 and cancer, this study provides valuable insights to the scientific community. These insights are instrumental in guiding the identification of novel targets and advancing the development of effective therapeutic strategies for treating cancer.

Q3. It should be of interest to understand and depict the possible relationship among Rheb and Ras and Rho. Indeed, there should be a relationship among these signal transducers, and it is clear (or it is possible) that the therapeutic approaches suggested for blocking Rheb could affect the other transducers mentioned.

Response) Thank you for your nice comment, we added information as below

Location: line 753-782

“To explore therapeutic strategies, we aim to elucidate the interplay among GTPase transducers, including Rho, Ras, and Rheb. This investigation aims to uncover additional avenues for inhibiting Rheb effectors, enhancing our understanding of potential therapeu-tic interventions.

Concerning the upstream signals involved in the crosstalk between Ras and Rho, it's noteworthy that both Ras and Rho can be jointly regulated by two Guanine Nucleotide Exchange Factors (GEFs): Sos and Ras-GRF [250]. Heptahelical receptors can similarly induce the simultaneous activation of Ras and Rho GTPases. Notably, this process is facil-itated by another superfamily of GTPases, known as heterotrimeric G proteins [250]. Re-garding the downstream signals of Ras and Rho, both Ras and Rho activate PI3K/Akt signaling to promote cell growth. This activation leads to an increase in the expression of genes involved in protein synthesis [251]. While Ras induces p21 through the ERK path-way, Rho inhibits p21 during G1 progression. [250]. Indeed, Sos has the ability to stimu-late cell migration by activating the Ras-Rho pathway [252]. In Ras-Rho pathway, Ras can activate Rho through the activation of Phosphoinositide 3-kinases (PI3K) [252]

When it comes to Ras and Rheb, Ras receives signals from proteins like Sos or Ras-GRF. However, Rheb doesn't necessarily require a dedicated GEF since its basal nucleotide exchange rates are sufficiently high to load it with GTP effectively [253]. Regarding the downstream effects of Ras and Rheb, Ras can hinder the TSC1/TSC2 complex through PI3K activation. This inhibition of the TSC complex by Ras-GTP enables Rheb to activate mTOR, the master regulator of translation [254,255]. In addition to the effect of Ras on Rheb activation, Rheb is activated by NOX2. Upon activation, NOX2 produces superoxide, which is then converted to hydrogen peroxide, likely by superoxide dismutase. Hydrogen peroxide inhibits TSC1/2, and when TSC1/2 is suppressed, it increases the levels of RheB-GTP. This, in turn, enhances the activity of mTORC1 [256].

As of now, there are no reports on the crosstalk between Rho and Rheb. However, it's worth noting that both Rheb and Rho could potentially act as downstream effectors of Ras and be regulated by Ras through the Akt-PI3K pathway. It is suggested that Ras inhibitors might be indirect Rheb inhibitors such as farnesyltransferase inhibitors.”

Q4. Actually, FTI (and GTI) inhibitors can have different target molecules other than Rheb. In other words, the authors should take into account how much the function of a given transducer is dependent on or independent of the others.  Also, the possibility that one transducer can substitute the other. This should be considered in detail with therapeutic options proposed.

Response:  Thank you for your great comment. We added more information to clarify the effect of FTI inhibitors on Rheb and other transducer as below:

Line 825-826:

Noticeably, lonafarnib at a concentration of 5μM did not inhibit Ras-GTP and its down-stream effectors such as MAPK and Akt [264].

Line 830-831:

Besides, Tipifarnib significantly inhibited HRas farnesylation in head and neck squamous cell carcinomas [267] and HRas in rhabdomyosarcoma [268].

Line 841-842: Additionally, FTI-277 inhibited breast cell invasion and migration by blocking HRas acti-vation [270]

Line 843-851:

FTIs might inhibit the post-translational lipid modification of HRas. While KRas and NRas are also substrates of farnesyl transferase, FTIs failed to prevent prenylation of K- and N-Ras. Compellingly, FTIs still inhibited the growth of mouse mammary tumor vi-rus (MMTV)-KRas and MMTV-NRas tumors in preclinical models. Besides, research on the molecular biology of the defective pathway has primarily concentrated on the activa-tion of H-Ras. However, the activation of K-Ras or other farnesylated proteins may be more critical in tumorigenesis [271]. This suggests that the anti-proliferative action of FTIs is dependent on blocking the farnesylation of other proteins such as Rheb [272]

Q5. Some references, in which Rheb is mentioned, should be considered and discussed. Alternatively, the author should explain why they do not take them into account.

NADPH Oxidase 2-Derived Reactive Oxygen Species Promote CD8+ T Cell Effector Function. Chen J, Liu C, Chernatynskaya AV, Newby B, Brusko TM, Xu Y, Barra JM, Morgan N, Santarlas C, Reeves WH, Tse HM, Leiding JW, Mathews CE.

J Immunol. 2023 Dec 11;212(2):258-70. doi: 10.4049/jimmunol.2200691. Online ahead of print.

PMID: 38079221   

Statins markedly potentiate aminopeptidase inhibitor activity against (drug-resistant) human acute myeloid leukemia cells. Jansen G, Al M, Assaraf YG, Kammerer S, van Meerloo J, Ossenkoppele GJ, Cloos J, Peters GJ. Cancer Drug Resist. 2023 Jul 4;6(3):430-446. doi: 10.20517/cdr.2023.20. eCollection 2023. PMID: 37842233

Response: Thank you for your nice recommendation. We added more information of each reference and discussed it as below:

  1. Chen, J.; Liu, C.; Chernatynskaya, A.V.; Newby, B.; Brusko, T.M.; Xu, Y.; Barra, J.M.; Morgan, N.; Santarlas, C.; Reeves, W.H.; et al. NADPH Oxidase 2-Derived Reactive Oxygen Species Promote CD8+ T Cell Effector Function. J Immunol 2024, 212, 258-270. https://doi:10.4049/jimmunol.2200691.

Location: line 874-876:

In addition to the effect of Ras on Rheb activation, Rheb is activated by NOX2. Upon activation, NOX2 produces superoxide, which is then converted to hydrogen peroxide, likely by superoxide dismutase. Hydrogen peroxide inhibits TSC1/2, and when TSC1/2 is suppressed, it increases the levels of RheB-GTP. This, in turn, enhances the activity of mTORC1 [256]

Location: line 774-775:

In addition, NOX2 inhibitors might be a potential option for blocking Rheb1-GTP via quenching reactive oxygen species (H2O2).

  1. Jansen, G.; Al, M.; Assaraf, Y.G.; Kammerer, S.; van Meerloo, J.; Ossenkoppele, G.J.; Cloos, J.; Peters, G.J. Statins markedly potentiate aminopeptidase inhibitor activity against (drug-resistant) human acute myeloid leukemia cells. Cancer Drug Resist 2023, 6, 430-446. https://doi:10.20517/cdr.2023.20

Location: line 853-855:

In a recent study, although Simvastatin is HMG-CoA reductase inhibitors, Simvastatin can impair the prenylation of Rheb in U937/WT, U937/CHR2863R0.2 and U937/CHR2863R5 cells [273].

Location: line 872-874:

Besides, a recent study showed Simvastatin, HMG-CoA reductase inhibitor, suppress prenylation of Rheb. It is suggested that other HMG-CoA reductase inhibitors such as Lovastatin and Pravastatin might inhibit the prenylation of Rheb.

Reviewer 3 Report

Comments and Suggestions for Authors

In their paper entitled “Role of Rheb1 and Rheb2 in Cancer development and Therapeutic Prospects“, the authors report on the current state of research into Rheb1 and Rheb2, from their biological characterisation to their involvement in cancer development. Therapeutic applications and inhibitors of their function are also described. The work is very well written and based on the current literature. There is not much to comment on. However, I suggest correcting the abbreviation on page 20: (CSVM! CSVL).

Author Response

Comments and Suggestions for Authors

In their paper entitled “Role of Rheb1 and Rheb2 in Cancer development and Therapeutic Prospects“, the authors report on the current state of research into Rheb1 and Rheb2, from their biological characterisation to their involvement in cancer development. Therapeutic applications and inhibitors of their function are also described. The work is very well written and based on the current literature. There is not much to comment on.

1) However, I suggest correcting the abbreviation on page 20: (CSVM! CSVL).

Response) Thank you for your nice comment, we corrected the abbreviation as below

Location: line 885-887

“Moreover, lonafarnib's ability to enhance tamoxifen and taxane-induced apoptosis is nul-lified by a mutant geranylgeranylated Rheb1 protein, Rheb1-CSVL, which has been de-rived from Rheb1-CSVM.”

Round 2

Reviewer 1 Report

Comments and Suggestions for Authors

I suggest acceptance of the manuscript in the present form.